# A Cryogenic 8-Bit 32 MS/s SAR ADC Operating Down to 4.2 K

**Yajie Huang** [1,2] , **Chao Luo** [1,2,*] and **Guoping Guo** [1,2]

1    The Physics Department, The University of Science and Technology of China, Hefei 230026, China
2    The Key Laboratory of Quantum Information, CAS, Hefei 230026, China
*    Correspondence: lc0121@ustc.edu.cn

**Abstract:** This paper presents a cryogenic 8-bit 32 MS/s successive approximation register (SAR) analog-to-digital converter (ADC) which operates down to 4.2 K. This work uses a modified liquid helium temperature (LHT) SMIC 0.18 μm CMOS technology to support the post-layout simulation. The proposed architecture adopts an offset-promoted dynamic comparator, waveform shaping circuit and true single-phase clock (TSPC) based sar logic circuit to achieve high realizing frequency and low power dissipation. At 1.8-V supply, 1.7 V input amplitude and 32 MS/s sampling frequency, the ADC achieves a power consumption of 2.4 mW and a signal-to-noise and distortion ratio (SNDR) of 47.7 dB, obtaining a figure of merit (FOM) of 378 fJ/conversion-step. The layout area of the ADC is about 0.253 mm$^2$.

**Keywords:** cryogenic CMOS; characterizaton; SAR ADC; quantum computer; low power

## 1. Introduction

Relevant research shows that quantum computers are overwhelming faster than conventional supercomputers in solving specific problems [1]. The researchers from Google developed a programmable superconducting quantum processor "Sycamore" containing 53 available qubits in 2019. The processor can sample an instance of quantum circuit for 1 million times in 200 s, which has incomparable advantages over classical supercomputers for the same task [2]. Additionally, a state-of-art quantum processor named "zuchongzhi 2.1" made the random quantum circuit sampling with a system scale up to 60 qubits and 24 cycles. The classical computational cost of simulating the state-of-art quantum processor's sample task is estimated to be 6 orders of magnitude higher than that of the most complex tasks on the "Sycamore" [3]. The overall power dissipation of the quantum computer system is about tens of kilowatts, which is only a few hundredths of that of classical supercomputers. Although the quantum computer has shown great computing potential in specific problems, it also has some difficulties such as the readout and control of quantum bits. Typically, the quantum processor operates down to cryogenic temperature to maintain the fidelity of quantum bits. The state-of-art quantum computers possess only a few qubits which can be controlled and read out by room-temperature (RT) electronics which are connected to the cryogenic qubits through only a few coaxial cables [4]. For example, the readout and control module of a 72-qubit superconducting quantum processor requires 168 long and lossy coax paths from 300 K to 4 K and 168 superconducting coax from 4 K to 10 mK [5]. Nevertheless, a practical quantum computer needs more than thousands of qubits, which means that this method is impractical due to the complexity and reliability of the system [6].

Therefore, making the readout and control circuits operate close to the quantum processor is a reasonable approach to solving the problems [7]. As illustrated in Figure 1, the state-of-art quantum processor control system is composed of two parts working at 300 K and 1–4 K, respectively. The 300 K part consists of ADC and control logic and the 1–4 K part consists of the mixer, digital-to-analog converter (DAC), low-noise amplifier

(LNA), high-frequency clock generator, and so on [8]. The proposed cryogenic control system places ADC and control logic at 4.2 K to minimize the coax between RT and LHT.

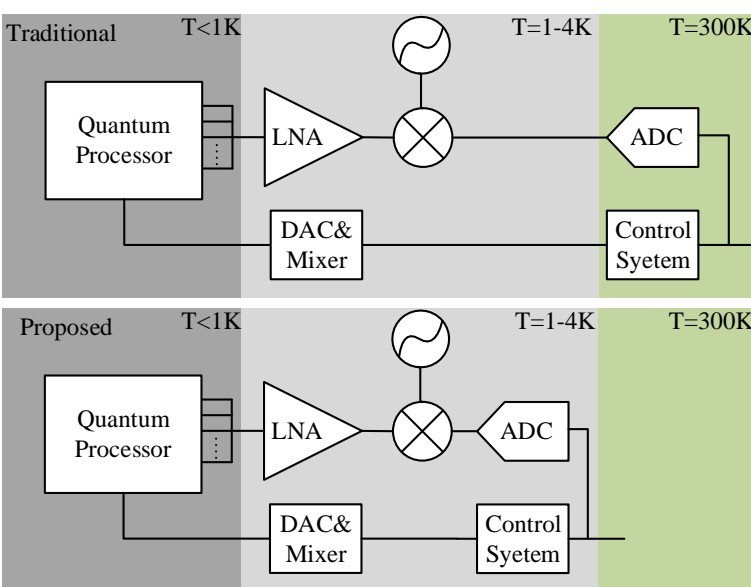

**Figure 1.** The traditional quantum processor control system and proposed cryogenic control system.

The most commonly used ADCs are Flash ADC, Pipeline ADC, Sigma-Delta ADC and SAR ADC [9]. The first three kinds of ADC have larger circuit scales and higher power dissipation compared with SAR ADC under the same performance [10–12]. However, power consumption is one of the most important constraints in low temperature circuit design due to the limit of the refrigerating power of the dilution refrigerator. Hence, most of the cryogenic ADC papers employ SAR architecture to meet the requirements of low power consumption and high frequency. The first cryogenic SAR ADC operating from 4 K to 300 K with 3 kHz sampling frequency which was fabricated in a 0.7 μm CMOS technology was reported in [13]. A time-interleaved SAR ADC in 40 nm CMOS process was introduced in [14], achieving effective number of bits (ENOB) of 6-bit at 4.2 K with sampling rate 1 GS/s. However, all above designs are based on the RT spice models rather than the LHT modified model. Since this design is the first time to use the compact LHT SMIC 0.18 μm CMOS spice model, the indexes of the proposed ADC are set to a relatively conservative range. Furthermore, the proposed ADC can be used as a sub-ADC to form a hybrid ADC to achieve higher speed or accuracy. This work has a solid foundation for the subsequent development of quantum chip readout and control circuits.

In the following, Section 2 introduces the measurement results and characterization of the cryogenic CMOS transistors. Section 3 describes the proposed SAR ADC architecture, the design and analysis of critical circuits. Section 4 summarizes the post-layout simulation results of the ADC.

## 2. Cryogenic CMOS Characterization

The performances of CMOS transistors and circuits are generally improved at cryogenic temperatures [15]. In general, as the temperature decreases, transistors exhibit many excellent electrical characteristics such as higher carrier mobility, higher saturation speed, better switching performance (lower subthreshold swing), vanishment of latch effect, lower power consumption, lower leakage current, lower thermal noise, better thermal conductivity and so on [16]. However, low-temperature CMOS also faces a series of problems, such as impurity freeze-out effect, kink effect, transient current, carrier mobility change, etc. [17]. These unique device characteristics at low temperature make it very difficult to describe the phenomenon and modify the RT model.

When the transistors produced in thin-oxide SMIC 0.18 μm CMOS technology are put into the 4.2 K liquid helium dewar, many of their characteristics will change greatly. The spice model of normal temperature process-design-kit (PDK) can not predict the behaviors of the devices at 4.2 K anymore [18]. Firstly, the threshold voltage $V_{th}$ increases apparently with the decrease of temperature due to the rise of Fermi potential and the freeze-out effect. According to the test results, the threshold voltage of NMOS (10 μm/180 nm) and PMOS (10 μm/180 nm) increases by more than 100 mV and 300 mV at LHT respectively. As a result, under the premise of the same power budget, the operating frequency of circuits, especially digital circuits will decrease at low temperatures. For high operating frequency cryogenic circuits, employing the body effect to reduce $V_{th}$ and using low threshold voltage transistors are reasonable approaches. Besides, the peak transconductance value ($G_{m-peak}$) shows a sharper change at 4.2 K than at 300 K which is presented in Figure 2. The reason for the sudden increase of $G_{m-peak}$ is that the ionized impurity scattering and lattice scattering will be weakened at cryogenic temperature resulting in the rise of carrier mobility ($\mu$) [19]. Assuming that $V_{DS}$ is constant (50 mV, e.g.), the relationship between carrier mobility and $G_{m-peak}$ is:

$$\mu_0 = \frac{G_{m-peak}L}{C_{OX}V_{DS}W} \tag{1}$$

where $C_{OX}$ represents the gate oxide capacitance. Generally, the ratio of $\mu_n$ to $\mu_p$ is 3:1 at RT, but at LHT it rises to 6:1. Furthermore, the transconductance is more sensitive to the change of $V_{GS}$ at LHT which will worsen the linearity of analog circuits and aggravate the total harmonic distortion [20], and the mismatch coefficients $A_{VT}$ and $A_\beta$ of transistors also increase at 4.2 K.

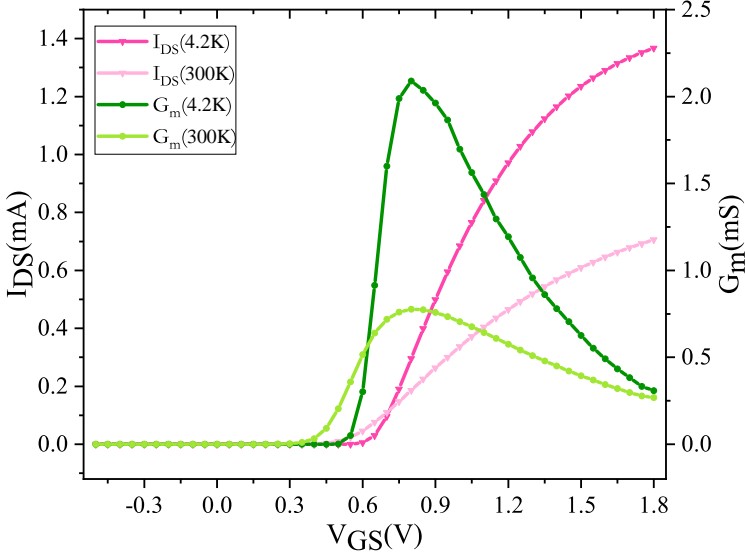

**Figure 2.** The difference of $G_m$ and $I_{DS}$ at RT and LHT.

Another cryogenic temperature characteristic of MOS transistors that cannot be ignored is the KINK effect. As illustrated in Figure 3a, when the bulk silicon MOSFET enters the saturation region and $V_{DS}$ is much greater than $V_{OD}$, the $I_{DS}$ appears a significant increase. The reason for this phenomenon is the freezing effect of the substrate and the collision ionization of the carrier. Due to these effects, holes accumulate in the substrate and the substrate potential $V_{BULK}$ increases[21]. The modified threshold voltage can be expressed as:

$$V_{TH\_KINK} = V_{FB} + 2\phi_F + \gamma\sqrt{2\phi_F - V_{BULK}} \tag{2}$$

where $V_{FB}$ refers to the flat band voltage, $\phi_F$ means the Fermi potential and $\gamma$ represents the substrate bias effect coefficient. The rise of $V_{BULK}$ leads to the decrease of $V_{TH\_KINK}$ and the increase of $I_{DS}$ so that the kink phenomenon emerges [21]. As shown in Figure 3b,

the difference between simulation results based on the revised spice model and the 4.2 K measurement results is tiny. It is worth mentioning that with the progress of the chip process, the kink effect gradually disappears with the decrease of circuit operating voltage. As for the processes above 160 nm, increasing substrate bias and reducing substrate doping concentration are effective to alleviate the kink effect.

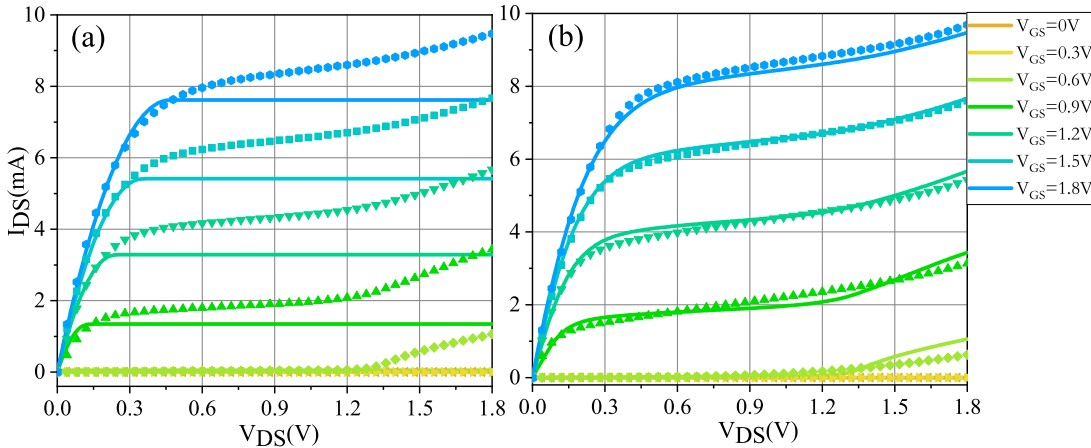

**Figure 3.** Spice model fitting, (**a**) kink effect, (**b**) the modified spice model.

## 3. Cryogenic SAR ADC

### 3.1. ADC Architecture

Figure 4 illustrates the detailed diagram of the proposed cryogenic SAR ADC. Fundamentally, it comprises a double-tail dynamic comparator, bootstrap sample-and-hole (S/H) circuit, capacitor array and successive approximation registers. To achieve lower power consumption and better linearity, this design employs a monotonic switching procedure [22,23]. The capacitor array is binary-weighted [24]. The CLK_in is generated by an off-chip sine wave signal which will be converted into a square wave through a wave shaping circuit and then passes through a frequency divider with a duty ratio of 1:4. The detail of the transform block is illustrated in Figure 5. Figure 5a is the schematic of the sinusoidal-to-square wave transform circuit. These resistors with the same resistance value ensure a 1:1 duty cycle and the two inverters convert the sine wave into a square wave. Figure 5b is the diagram of the one-fifth frequency divider. The sample clock of the proposed architecture is generated at the output node of the first stage TSPC.

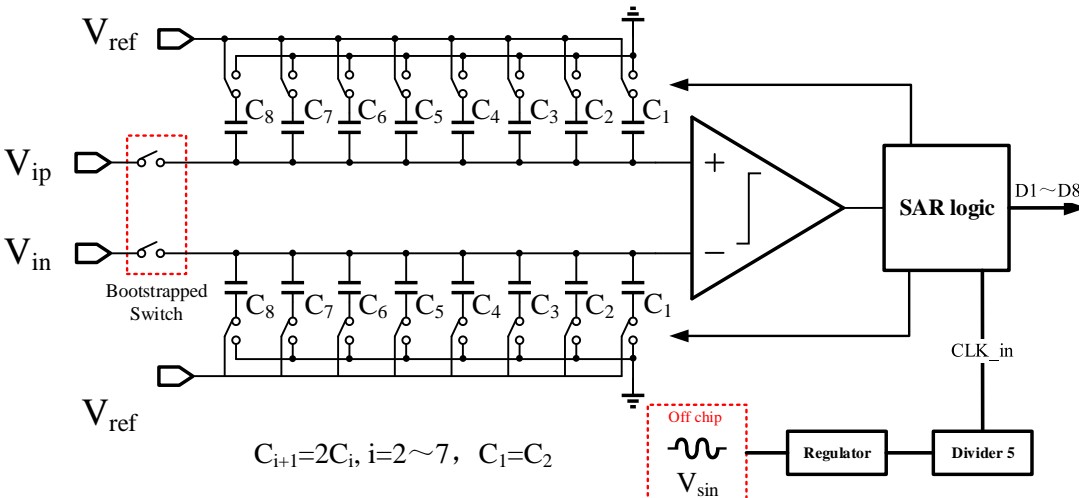

**Figure 4.** The proposed SAR ADC structure.

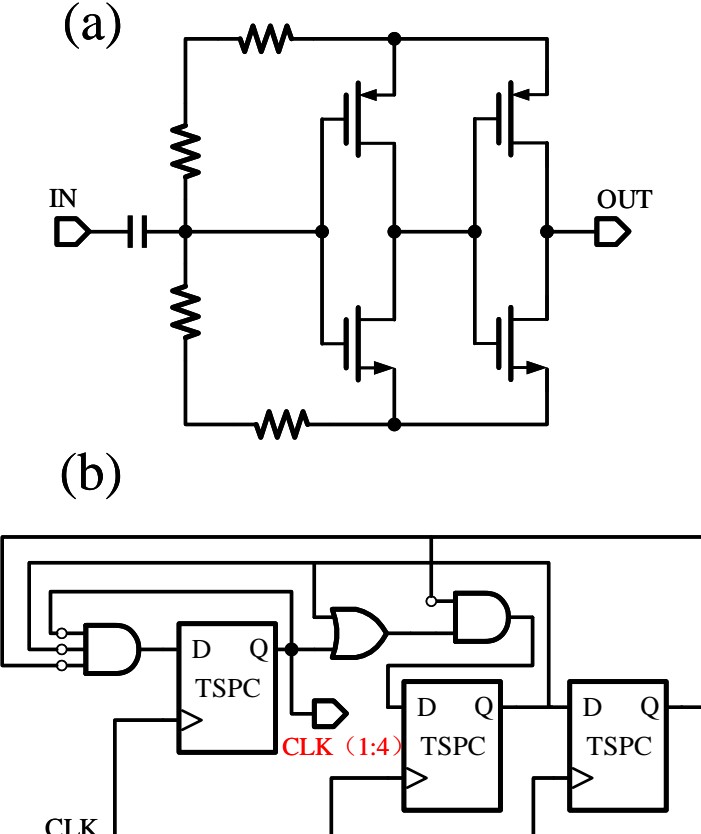

**Figure 5.** (**a**) The schematic of the sinusoidal-to-square wave transform circuit, (**b**) The diagram of the one-fifth frequency divider.

To repress the supply and substrate voltage noise, the fully differential architecture is adopted which also has good common-mode noise suppression capability. Because of the fully differential scheme, the operations of the two input nodes are complementary. Take the positive end operation of the ADC as an example. In the sampling stage, the top plate of the capacitor is charged to $V_{ip}$ via the bootstrapped switches, which is conducive to the establishment speed and input bandwidth [25]. Meanwhile, the bottom plates of the capacitor are reset to $V_{ref}$. Afterward, the bootstrapped switches are disconnected during the hold stage and the comparator begins the first comparison without any switching operation of the capacitor. According to the output of comparator, the maximum capacitor $C_8$ that has been charged to a higher voltage potential is connected to ground while the other one (possesses lower voltage potential) maintains the initial state.

The ADC repeats the process until the LSB is determined. Since there is only one capacitor switch per bit cycle which reduces the charge transfer in the capacitive DAC network and the conversion of the control circuit and the switch buffer, the ADC has less power dissipation [26]. Figure 6 illustrates the monotonic switching procedure by a 3-bit example. The every potential switching scenario of this switching procedure is displayed. The waveform of the switching procedure is shown in Figure 7. Compared with the conventional $V_{cm}$-based switching procedure, the $V_{cm}$ of the monotonic switching procedure is dropped from $1/2$ $V_{ref}$ to ground. This switching procedure scheme does not demand an upward transition. At liquid helium temperature, the on-resistance of an N-type MOSFET is about 1/6 that of a P-type MOSFET one at the same transistor size because the carriers mobility at 4.2 K is approximately twice that at 300 K [15]. Compared with the traditional capacitor array, this structure saves half the number of unit capacitors.

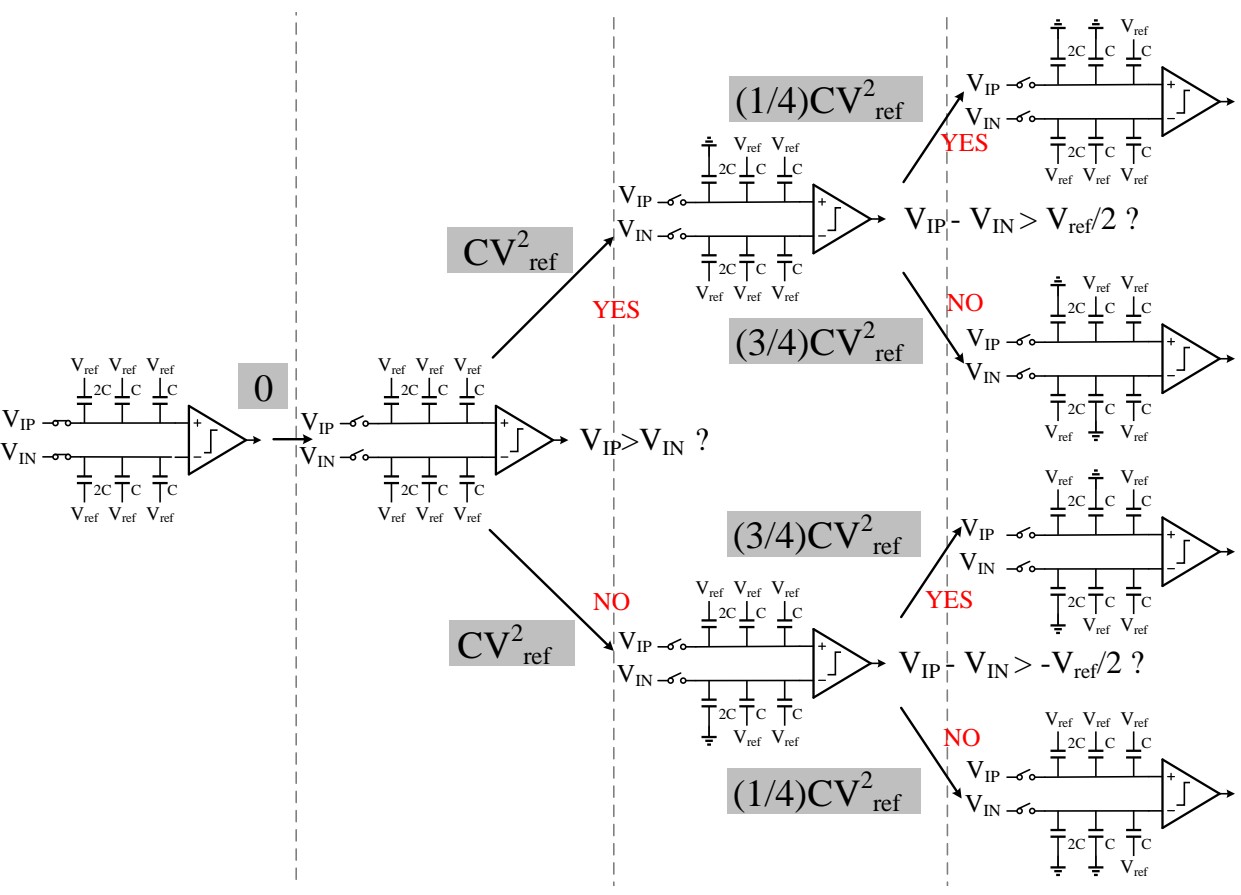

**Figure 6.** 3-bit monotonic switching procedure.

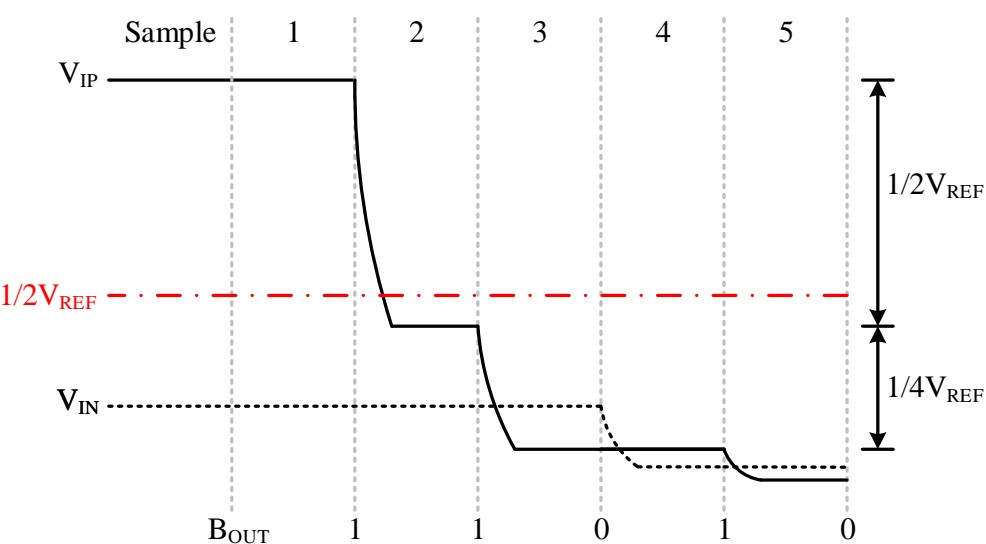

**Figure 7.** Waveform of monotonic switching procedure.

The average switching energy for an n-bit SAR ADC using this switching procedure can be derived as

$$E_{avg} = \sum_{i=1}^{N} 2^{N+1-2i} \left( 2^i - 1 \right) CV_{ref}^2 \tag{3}$$

for the 8-bit ADC, the monotonic switching procedure consumes only 63.5 $CV_{ref}^2$ about 1/5 of the conventional one. This architecture also saves the switchers, unit capacitors and digital control logic which leads to a lower hardware budget.

### 3.2. Dynamic Comparator

Figure 8 illutrates the transistor-level circuit of proposed dynamic comparator. In the conversion phase, the input voltage of the comparator is gradually approaches the ground. To function properly within the input common-mode voltage range from half $V_{ref}$ to ground, the comparator employs a P-type input pairs. When the clock control signal Clkc of the comparator is high voltage, node N and node P are discharged to GND, indicating that the comparator is in the reset stage [27]. On the contrary when Clkc goes to low level, node N and node P are charged at different charging speeds according to the electrical level of the input signal $V_{in}$ / $V_{ip}$, and the comparator is in comparison stage. Meanwhile, the outputs of the latching circuit are drawn to VDD at the reset stage. Once the preamplifier goes to the comparison stage, assuming node N charges faster, the M10 turns off first and the drain of M10 starts discharging through a parasitic capacitor where the drain of M7 maintains high level due to the positive feedback effect. Thus, the $OUT_p$ changes to high level and the $OUT_n$ keeps low. Eventually, the Valid signal genereded by the $OUT_p$ and $OUT_n$ passing through NAND gate turns to high level to enable asynchronous control logic.

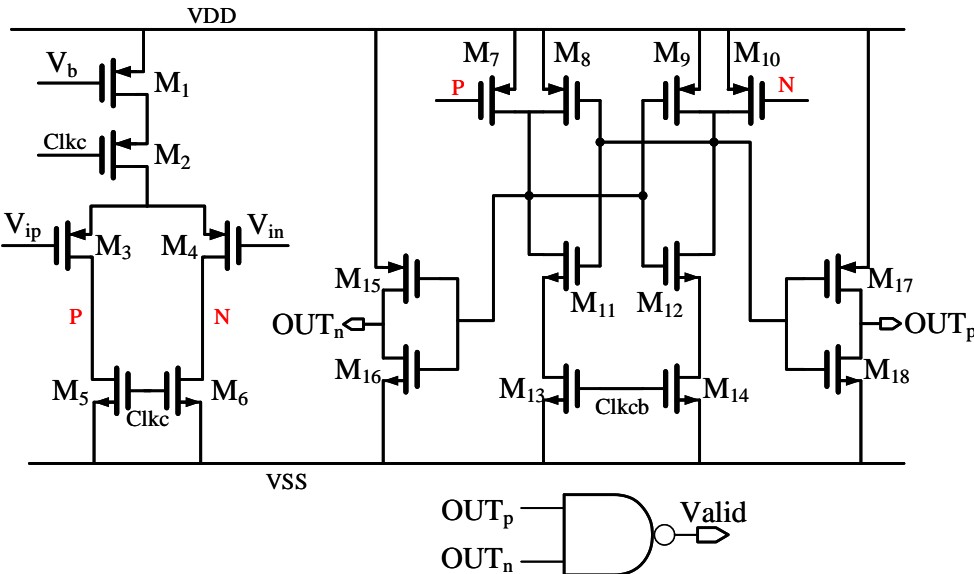

**Figure 8.** The proposed dynamic comparator circuit.

According to the schematic, the total input offset voltage can be expressed as

$$V_{OS} = \sqrt{V_{os\_pre}^2 + \frac{1}{A^2} V_{os\_latch}^2} \tag{4}$$

where the $V_{os\_pre}$ is the offset voltage of the preamplifier, $V_{os\_latch}$ is the offset of the latching circuit and A is the gain of the preamplifier. Obviously, the input offset voltage of the preamplifier plays a decisive role in the total offset voltage because the coefficient of $V_{os\_latch}$ is tiny (proportional to the reciprocal of the square of A). The square of the offset voltage of the preamplifier is

$$V_{OS,pre}^2 = \Delta V_{th3,4}^2 + \left( \frac{V_{GS3,4} - V_{th3,4}}{2} \right)^2 \left[ \left( \frac{\Delta C_{P,N}}{C_{P,N}} \right)^2 + \left( \frac{\Delta \beta_{3,4}}{\beta_{3,4}} \right)^2 \right] \tag{5}$$

where $\Delta V_{th3,4}$ is the threshold voltage offset of the differential pair $M_3$ and $M_4$, $\Delta C_{P,N}$ is the mismatch of the capacitance to ground of the nodes P and N, $\Delta \beta_{3,4}$ is the transistor current coefficient mismatch. Enlarging the transistor size can decrease the dynamic offset but it will increase the power dissipation and the chip area. The coefficients $\Delta C_{P,N}$ and $\Delta \beta_{3,4}$ are affected by the input common-mode voltage. This circuit introduces a biased PMOS $M_1$ to

cascode the switch PMOS $M_2$. $M_1$ is biased in the saturation region so that change of its $V_{DS}$ has just a small impact on the $I_{DS}$. Therefore, no matter how the input common-mode voltage changes, the overdrive voltage $V_{GS3,4} - V_{th3,4}$ remains near a constant value.

Moreover, the settling time of the preamplifier has been present in Equation (7) [28]. At cryogenic temperature, the threshold voltage of the used PMOS has increased by 26% and the $I_{DS3,4}$ has increased by 8.5%. Thus, the comparison time at LHT is longer than that at RT. The transient simulation results at 300 MHz clock of the comparator shown in Figure 9 indicate that the comparison time at LHT is 210 ps longer than that at RT. Therefore, the proposed dynamic comparator needs higher energy consumption to achieve the required performance at low temperature.

$$T_{P_{\text{preamp}}} \approx T_{\text{lnt}} \approx \frac{|V_{\text{THP}}| \cdot C_{FP}}{I_{DS3,4}} \tag{6}$$

**Figure 9.** The transient simulation results of the comparator at RT and LHT.

### 3.3. Bootstrapped Switch

Figure 10 is the transistor-level circuit diagram of the proposed bootstrap switch. When Clk is low, the bootstrap capacitor C3 is charged to VDD by M8, and the gate of switch transistor M1 is pulled down to GND by M6 and M7, the sampling capacitors enter the hold state and M3, M5 turn off to cut off the passway between C3 and M1 at the same time [29,30]. As Clk goes high, M3 and M12 turn off, M8 and M9 turn on, C3 is connected to the source and gate of M1 which ensures that the $V_{GS}$ of M1 equivalent to VDD is not affected by the change of input voltage. Meanwhile, M6 and M7 turn off, and the gate of M1 disconnects from GND. To keep the on-resistance of the sampling switch transistor constant, this circuit adds a PMOS M2 to compose a complementary switch. This structure can limit the change of the on-resistance. During the hold state, the electrons from the drain of NMOS and the holes from the source of PMOS will be canceled out. Consequently, the channel charge injection effect is suppressed to some extent. The dynamic performance of the proposed bootstrap switch with a 4 pF load is presented in Figure 11. The Nyquist-rate input post-layout simulation result shows an SNDR of 69.53 dB and an ENOB of 11.26 bits which far exceed the demand of an 8-bit SAR ADC.

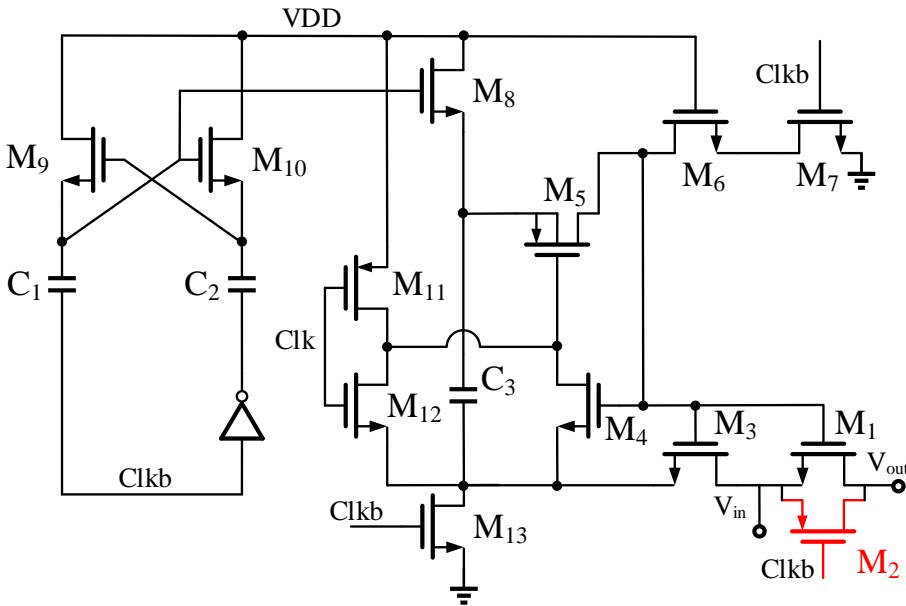

**Figure 10.** The proposed bootstrap switch circuit with complementary switch transistors.

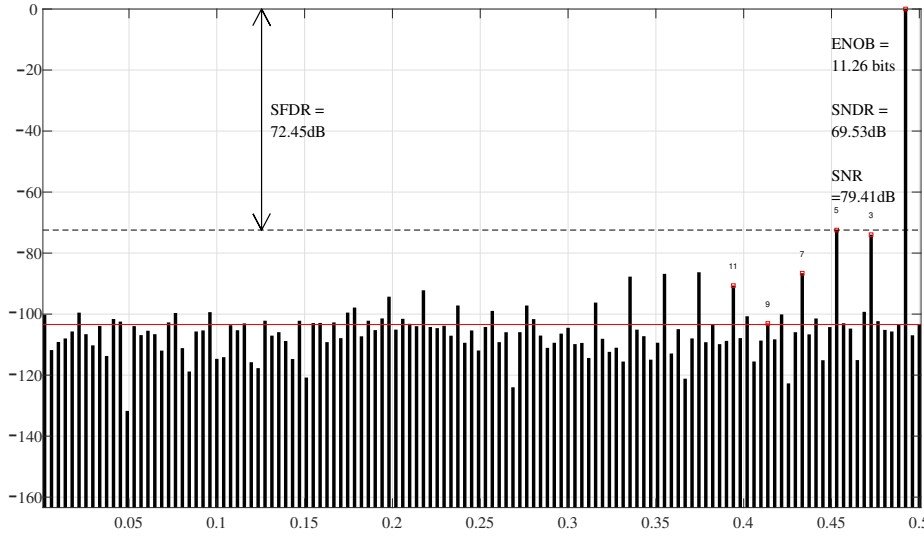

**Figure 11.** Simulated FFT spectrum at 32 MS/s with 15.7 MHz input of the bootstrapped switch

### 3.4. SAR Logic and Capacitor Array

To avoid the introduction of high-frequency clock generation circuit, this paper adopts an asynchronous clock control logic to generate an internal clock. Different from the traditional timing generation circuit, this design uses the TSPC structure D flip-flop to obtain higher clock frequency, lower power consumption and a smaller layout area. Through simulation and comparison, with the same ADC performance, the power dissipation of SAR logic part using TSPC structure (906 µW) is less than one-half of the DFF structure (1.9 mW) composed of NAND gate at LHT. Figure 12 shows the transistor-level circuit of the TSPC. Figures 13a and 14 are schematic diagrams of the asynchronous clock generation circuit and timing diagram respectively. Wherein, Clks is the control clock of the bootstrap switch, which is obtained from the sinusoidal signal outside the chip through the waveform conversion circuit, and the duty cycle is 1:4. The output signals of the comparator $OUT_p$ and $OUT_n$ generate valid signals through an XOR gate. Clks, valid and Clk8 generate Clkc through a three inputs NOR gate. This apporach avoids the introduction of an additional

high-frequency clock and the transmission attenuation of high frequency signals between RT and LHT.

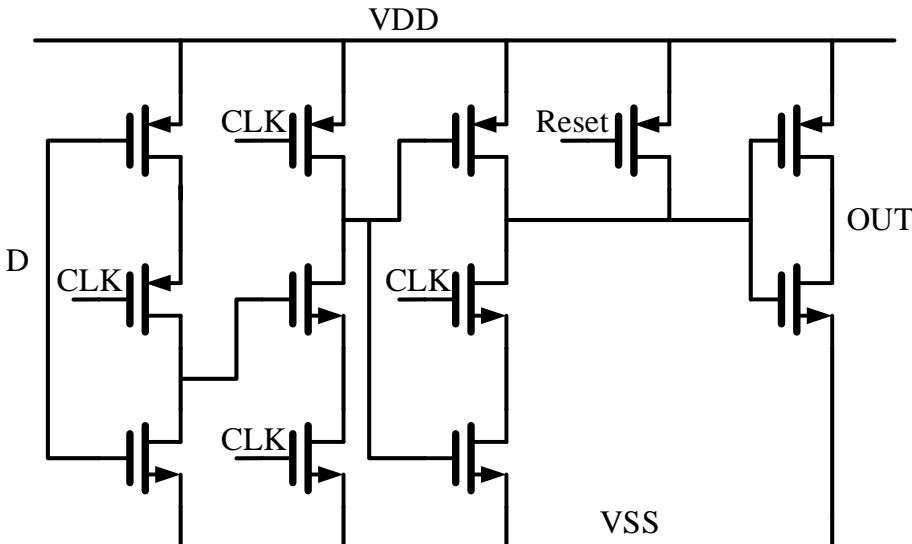

**Figure 12.** The schematic of the TSPC.

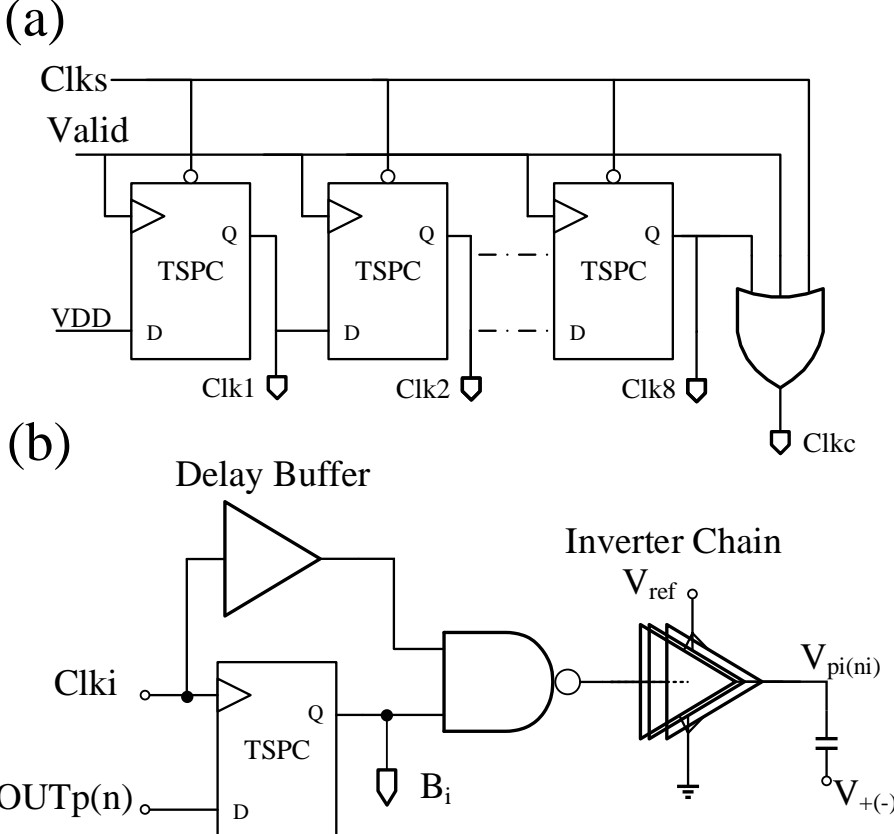

**Figure 13.** (**a**) Schematic of asynchronous control logic, (**b**) the control logic of DAC.

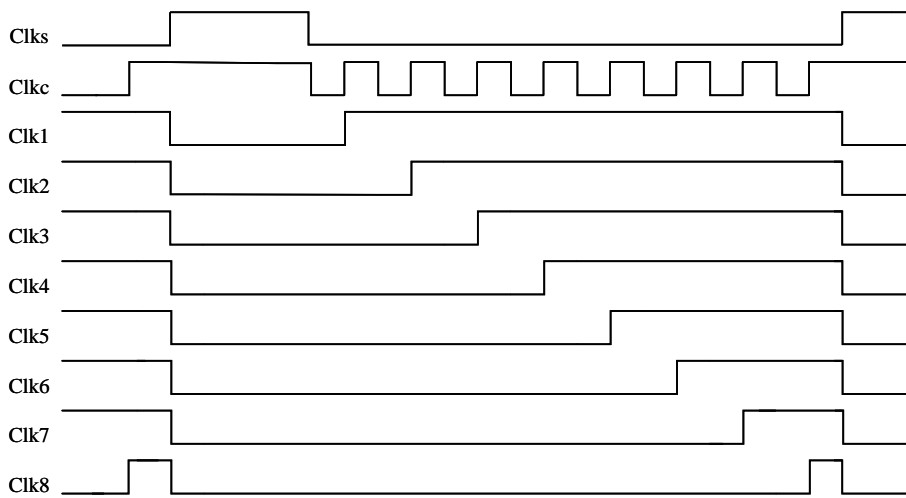

**Figure 14.** Timing diagram of asynchronous control logic.

As shown in Figure 13b, Clk1 to Clk8 are used as the clock of the level conversion circuit of the capacitor array. On the rising edge of Clki, the $OUT_{p/n}$ of the comparator is sent to TSPC. If the $OUT_{p/n}$ is at high level, the corresponding capacitor is changed from $V_{ref}$ to ground. When $OUT_{p/n}$ goes low, the associated capacitor is still linked to $V_{ref}$. At the falling edge of Clki, all capacitors are reconnected to $V_{ref}$. The inverter chain is used as a switch buffer to switch between $V_{ref}$ and ground. This work uses metal–insulator–metal (MIM) capacitors and the capacitance of the unit capacitor is 16 fF (4 μm × 4 μm). The total capacitance of the two capacitor networks is about 8.2 pF. The total active area of the networks is 580 μm × 310 μm about 71% of the whole ADC. Figure 15 illustrates the layout of the entire cryogenic SAR ADC core.

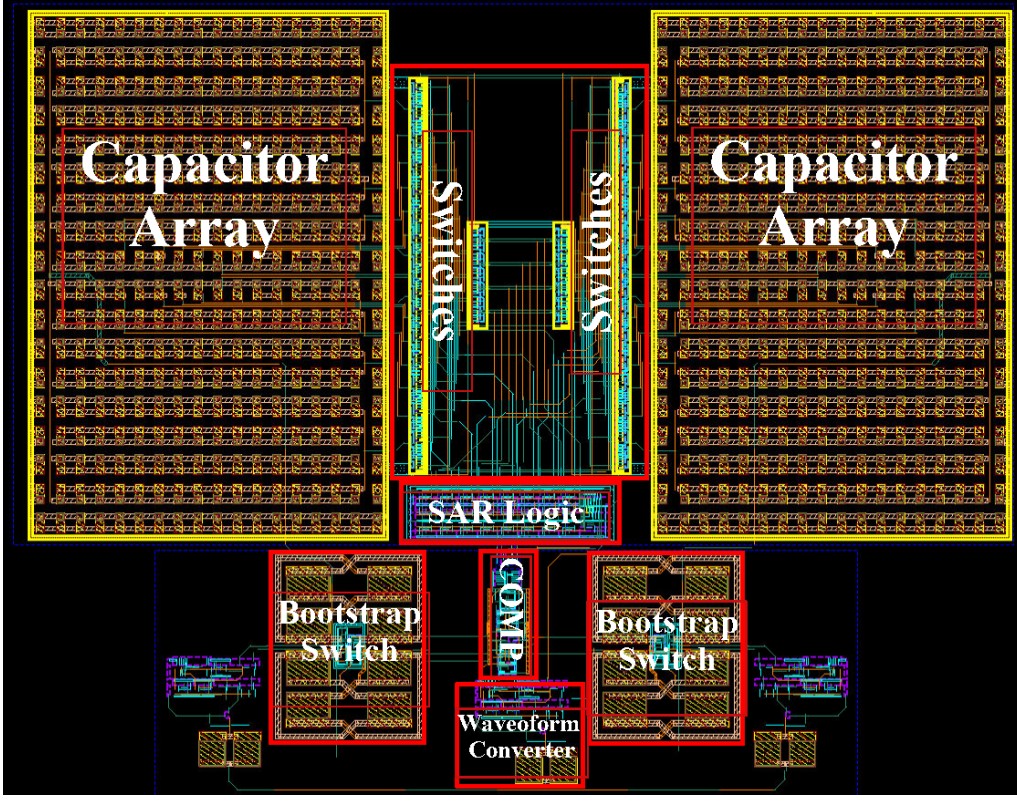

**Figure 15.** The layout of the proposed cryogenic SAR ADC.

## 4. Simulation Results

The post-layout simulation of the cryogenic SAR ADC is based on the modified SPICE model of the SMIC 0.18 μm CMOS technology which could fit the cryogenic performance of MOSFETs well. Figure 16 shows the simulated clock diagram of the proposed architecture. *Clk_in* is a 160 MHz off-chip sine wave input signal and *Clk_sq* is the output of the sinusoidal-to-square circuit. Then, *Clk_sq* is transformed to Clks by the one-fifth frequency divider. The establishment processes of valid and Clkc have been explained in Section 3. The dynamic performance of the ADC is simulated by analyzing a Fast Fourier Transform (FFT) of the output codes. It can be seen from Figure 17 that the peak SNDR and spurious free dynamic range (SFDR) for a 15.7 MHz sinewave input were measured at 47.7 dB and 54.53 dB with a clock frequency of 32 MHz, respectively. The static linearity of the ADC was simulated by analyzing over $6 \times 10^6$ output codes according to the histogram test approach [31]. Figures 18 and 19 show the DNL and INL versus output code. The peak differential nonlinearity (DNL) is $0.82/-0.98$ LSB and the peak integral nonlinearity (INL) is $1.1/-1.5$ LSB. Since the increase of transistors saturation current is caused by kink effect, the power dissipation of the ADC is up to 2.4 mW. The comparison between the proposed and other cryogenic SAR ADCs is summarized in Table 1. The proposed ADC offers a high resolution while having an input range of 1.7 V.

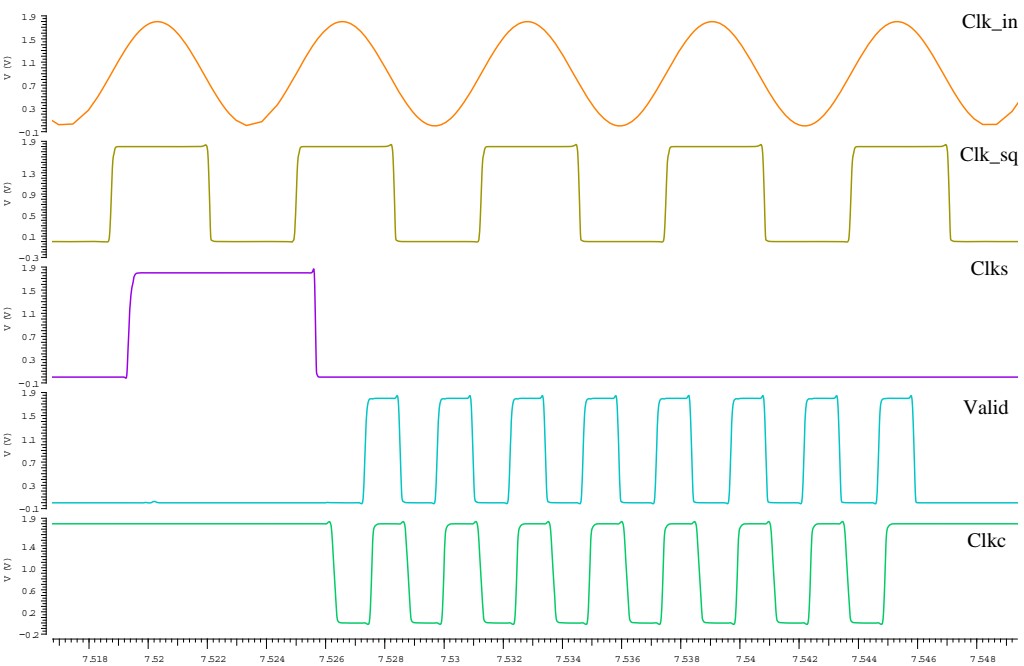

**Figure 16.** The simulated clock diagram of the proposed architecture.

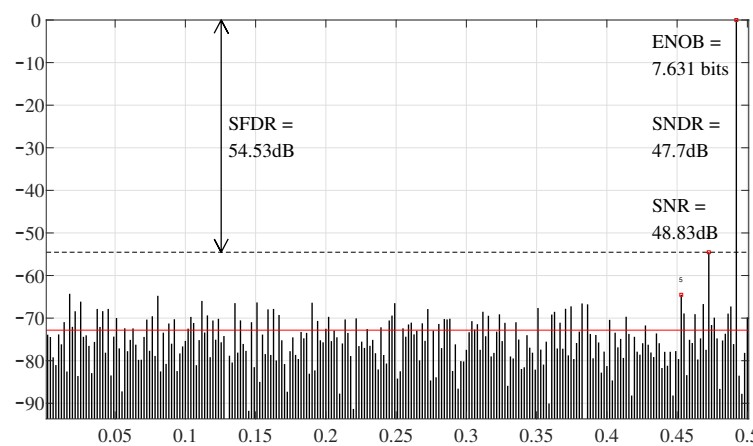

**Figure 17.** Simulated FFT spectrum at 32 MS/s with 15.7 MHz input of the ADC.

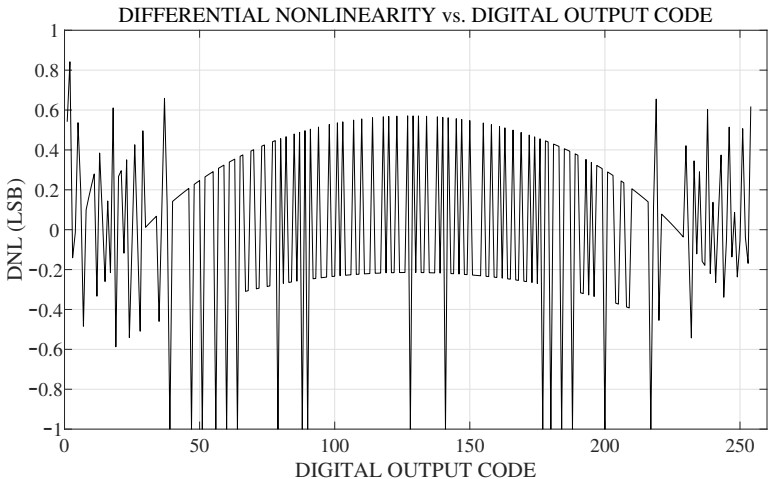

**Figure 18.** The DNL of the SAR ADC.

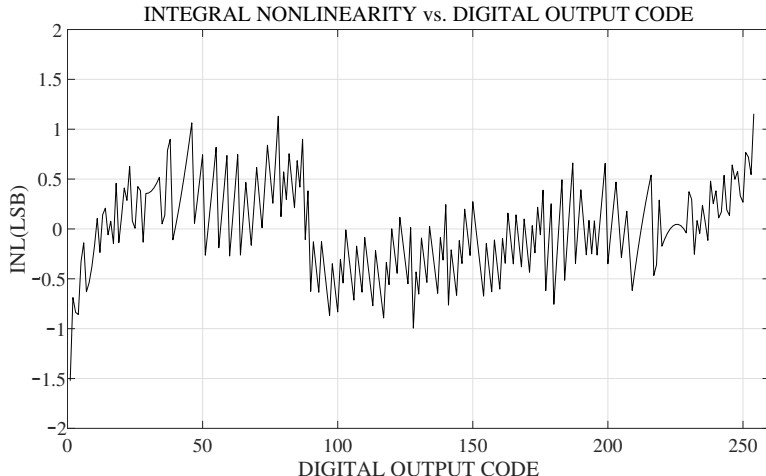

**Figure 19.** The INL of the SAR ADC.

**Table 1.** Performance summary and comparisions.

|  | This Work | [14] | [13] |
|---|---|---|---|
| Architecture | SAR | TI SAR | SAR |
| Technology | 180 nm | 40 nm | 0.7 μm |
| Sample frequency | 32 MS/s | 1 GS/s | 3 kS/s |
| Supply voltage | 1.8 V | 1.1 V | 5 V |
| Input range ($V_{pp}$) | 1.7 V | 0.7 V | 5 V |
| SNDR@Nyquist | 47.7 dB | 36.2 dB | - |
| SFDR@Nyquist | 54.53 dB | 48.5 dB | - |
| Power | 2.4 mW | 10.6 mW | 0.35 mW |
| FoM | 378 fJ/Conv.-step | 200 fJ/Conv.-step | - |

## 5. Conclusions

In this paper, a cryogenic SAR ADC operating at 4.2 K was presented. The proposed dynamic comparator and bootstrapping switch ensure that both power dissipation and signal linearity are taken into account at relatively fast speeds. The waveform shaping circuit and TSPC-based asynchronous control logic avoid the introduction of high frequency clock generation circuit. The prototype achieves a 32-MS/s operating speed and the power consumption is about 2.4 mW, which is affected by the kink effect at cryogenic temperature. The FOM of the design is 378 fJ per conversion-step (see the Appendix A). The post-simulation based on the LHT SPICE model demonstrates the practicality and application prospect of the design applied to the cryogenic quantum computer. In future work, the cryogenic model needs to be improved in terms of noise and mismatch and the design needs to be taped out.

**Author Contributions:** Research methodology, Y.H. and G.G.; circuit design, Y.H.; layout drawing, Y.H.; writing—original draft preparation, Y.H. and C.L.; supervision, G.G.; validation, Y.H. and C.L. All authors have read and agreed to the published version of the manuscript.

**Funding:** This research was funded by the National Natural Science Foundation of China under Grant No. 12034018.

**Data Availability Statement:** Data are contained within the article.The data presented in this study are available in this paper.

**Conflicts of Interest:** The authors declare no conflict of interest.

## Appendix A

The FOM [32] could be expressed as:

$$\text{FOM} = \frac{\text{Power}}{2^{\text{ENOB}} \cdot \min\{f_s, 2\text{ERBW}\}} \tag{A1}$$

In Equation (A1), fs represented sampling frequency , Power was the power consumption of the SAR ADC, ERBW was the effective resolution bandwidth and the effective number of bits of the ADC was represented by ENOB.

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
