# Peer review of "A Cryogenic 8-Bit 32 MS/s SAR ADC Operating down to 4.2 K"

_electronics, doi:10.3390/electronics12061420_

Round 1
Reviewer 1 Report
This work proposes a solution for a cryogenic SAR-DAC using a modified liquid helium temperature SMTC and achieves higher frequency operation and better circuit performances in quantum processing.
Please consider language and grammar revision.
In the introduction, give a description of the novelty of your work, and place it in relation with relevant literature.
Sections 2 and 3 are well written and well explained.
The simulation results section should be extended with relevant simulation for all the topics covered in section 3. Also compare your results with other relevant from literature.
Define the figure of merit from table 1.
Was the chip also fabricated?
Reviewer 2 Report
The overall presentation of this work must be improved. Special attention is required on format. Figures are not properly included and commented on such as Figure 1, Figure 4 etc. Results must be improved.
Reviewer 3 Report
This paper presents an 8-bit 32MS/s cryogenic register (SAR) 1 for successive approximation.An analog-to-digital converter (ADC) operates upto 4.2K. In this work, modified liquid 2-helium temperature (LHT) SMIC 0.18 µm CMOS technology is used to assist post-simulation work. The proposed 3 architectures use offset-assisted dynamic comparators, waveform shaping circuits, and 4 TSPC-based SAR logic circuits to achieve high operating frequencies and low power consumption. The authors claim that the proposed dynamic comparator and bootstrap switch ensure that the power consumption and signal linearity are calculated at a relatively fast rate and The TSPC-based asynchronous control logic and waveform shaping circuit avoids the introduction of high frequency clocking circuitry. Could the authors give any physical reason to happen this? Also can they illustrate this comparing with recent publications? Could some parts of the claims be validated experimentally? The paper is having high similarity. The authors are required to remove that.
Reviewer 4 Report
Dear Authors,
please, refer to the attached document. Thanks!

Reviewer 5 Report
The paper describes the results of implementing a SAR-ADC to operate at cryogenic temperature. The concept is to bring the SAR-ADC along other circuits to operate at 1-4K for cryogenic computers. It provides the circuit description, layout and simulations. Nevertheless, I would like to suggest some improvements:
* Please add the meaning of all acronyms to their firts apearance on the main text, even if they were presented in the abstract.
* The Introduction needs major improvements. Please describe clearly the goal of the article.
* The simulations presented are post-layout simulations?
Round 2
Reviewer 2 Report
The comments are not properly addressed.
Reviewer 3 Report
The answers are satisfactory
Round 3
Reviewer 2 Report
Paper can be accepted.
Author Response
Dear reviewer:
Thank you very much for your kind comments on our manuscript. There is no doubt that these comments are valuable and very helpful for revising and improving our manuscript.
Thank you again for your positive and constructive comments and suggestions on our manuscript.
We hope you will find our revised manuscript acceptable for publication.